# SMASH: One-Shot Model Architecture Search through HyperNetworks

**Andrew Brock, Theodore Lim, & J.M. Ritchie**
School of Engineering and Physical Sciences
Heriot-Watt University
Edinburgh, UK
{ajb5, t.lim, j.m.ritchie}@hw.ac.uk

**Nick Weston**
Renishaw plc
Research Ave, North
Edinburgh, UK
Nick.Weston@renishaw.com

## Abstract

Designing architectures for deep neural networks requires expert knowledge and substantial computation time. We propose a technique to accelerate architecture selection by learning an auxiliary HyperNet that generates the weights of a main model conditioned on that model's architecture. By comparing the relative validation performance of networks with HyperNet-generated weights, we can effectively search over a wide range of architectures at the cost of a single training run. To facilitate this search, we develop a flexible mechanism based on memory read-writes that allows us to define a wide range of network connectivity patterns, with ResNet, DenseNet, and FractalNet blocks as special cases. We validate our method (SMASH) on CIFAR-10 and CIFAR-100, STL-10, ModelNet10, and Imagenet32x32, achieving competitive performance with similarly-sized hand-designed networks.

## 1 Introduction

The high performance of deep neural nets is tempered by the cost of extensive engineering and validation to find the best architecture for a given problem. High-level design decisions such as depth, units per layer, and layer connectivity are not always obvious, and the success of models such as Inception (Szegedy et al., 2016), ResNets (He et al., 2016), FractalNets (Larsson et al., 2017) and DenseNets (Huang et al., 2017) demonstrates the benefits of intricate design patterns. Even with expert knowledge, determining which design elements to weave together requires ample experimentation.

In this work, we propose to bypass the expensive procedure of fully training candidate models by instead training an auxiliary model, a HyperNet (Ha et al., 2017), to dynamically generate the weights of a main model with variable architecture. Though these generated weights are worse than freely learned weights for a fixed architecture, we leverage the observation (Li et al., 2017) that the relative performance of different networks early in training (i.e. some distance from the eventual optimum) often provides a meaningful indication of performance at optimality. By comparing validation performance for a set of architectures using generated weights, we can approximately rank numerous architectures at the cost of a single training run.

To facilitate this search, we develop a flexible scheme based on memory read-writes that allows us to define a diverse range of architectures, with ResNets, DenseNets, and FractalNets as special cases. We validate our one-Shot Model Architecture Search through HyperNetworks (SMASH) for Convolutional Neural Networks (CNN) on CIFAR-10 and CIFAR-100 (Krizhevsky and Hinton, 2009), Imagenet32x32 (Chrabaszcz et al., 2017), ModelNet10 (Wu et al., 2015), and STL-10 (Coates et al., 2011), achieving competitive performance with similarly-sized hand-designed networks.

## 2 Related Work

Modern practical methods for optimizing hyperparameters rely on random search (Bergstra and Bengio, 2012) or Bayesian Optimization (BO) (Snoek et al., 2012; 2015), treating the model performance

as a black box. While successful, these methods require multiple training runs for evaluation (even when starting with a good initial model) and, in the case of BO, are not typically used to specify variable-length settings such as the connectivity and structure of the model under consideration. Relatedly, bandit-based methods (Li et al., 2017) provide a framework for efficiently exploring the hyperparameter space by employing an adaptive early-stopping strategy, allocating more resources to models which show promise early in training.

Evolutionary techniques (Floreano et al.; Stanley et al.; Suganuma et al., 2017; Wierstra et al., 2005) offer a flexible approach for discovering variegated models from trivial initial conditions, but often struggle to scale to deep neural nets where the search space is vast, even with enormous compute (Real et al., 2017).

Reinforcement learning methods (Baker et al., 2017; Zoph and Le, 2017) have been used to train an agent to generate network definitions using policy gradients. These methods start from trivial architectures and discover models that achieve very high performance, but can require twelve to fifteen *thousand* full training runs to arrive at a solution.

The method that most resembles our own is that of Saxe et al. (Saxe et al., 2011), who propose to efficiently explore various architectures by training only the output layer of convolutional networks with random convolutional weights. While more efficient than fully training an entire network end-to-end, this method does not appear to scale to deeper networks (Yosinski et al., 2014). Our method is conceptually similar, but replaces random weights with weights generated through HyperNets (Ha et al., 2017), which are one of a class of techniques for dynamically adapting weights through use of an auxiliary model (Denil et al., 2013; Jia et al., 2016; Rebuffi et al., 2017; Schmidhuber, 1992). In our case we learn a transform from a binary encoding of an architecture to the weight space, rather than learning to adapt weights based on the model input.

Our method is explicitly designed to evaluate a wide range of model configurations (in terms of connectivity patterns, depth, and width) but does not address other hyperparameters such as regularization, learning rate schedule, weight initialization, or data augmentation. Unlike the aforementioned evolutionary or RL methods, we explore a somewhat pre-defined design space, rather than starting with a trivial model and designating a set of available network elements. While we still consider a rich set of architectures, our method cannot discover wholly new structures on its own and is constrained in that it only dynamically generates a specific subset of the model parameters. Additionally, although our method is not evolutionary, our encoding scheme is reminiscent of CGP (Miller and Thomson, 2000).

Stochastic regularization techniques such as Dropout (Srivastava et al., 2012), Swapout (Singh et al., 2016), DropPath (Larsson et al., 2017) or stochastic depth (Huang et al., 2016) superficially resemble our method, in that they obtain variable configurations by randomly dropping connectivity paths in a fixed network architecture. Convolutional neural fabrics (Saxena and Verbeek, 2016), for example, leverage this idea to attempt to train one large network as an implicit ensemble of all subnetworks produced through dropping paths. A key element that sets our method apart is that the weights for each node in our network are dynamically generated, rather than fixed; if a Dropout ensemble were to visit a unit that had not previously been trained, the unit's weights would be completely untuned. Our method generalizes even to previously unseen architectures, and the network we train under stochastic conditions is merely a proxy we use to evaluate network configurations, rather than the final model.

## 3 One-Shot Model Architecture Search through HyperNetworks

In SMASH (Algorithm 1), our goal is to rank a set of neural network configurations relative to one another based on each configuration's validation performance, which we accomplish using weights generated by an auxiliary network. At each training step, we randomly sample a network architecture, generate the weights for that architecture using a HyperNet, and train the entire system end-to-end through backpropagation. When the model is finished training, we sample a number of random architectures and evaluate their performance on a validation set, using weights generated by the HyperNet. We then select the architecture with the best estimated validation performance and train its weights normally.

---

**Algorithm 1** SMASH

---

**input** Space of all candidate architectures, $\mathbb{R}_c$
   Initialize HyperNet weights $H$
   **loop**
      Sample input minibatch $x_i$, random architecture $c$ and architecture weights $W = H(c)$
      Get training error $E_t = f_c(W, x_i) = f_c(H(c), x_i)$, backprop and update $H$
   **end loop**
   **loop**
      Sample random $c$ and evaluate error on validation set $E_v = f_c(H(c), x_v)$
   **end loop**
   Fix architecture and train normally with freely-varying weights $W$

---

SMASH comprises two core components: the method by which we sample architectures, and the method by which we sample weights for a given architecture. For the former, we develop a novel "memory-bank" view of feed-forward networks that permits sampling complex, branching topologies, and encoding said topologies as binary vectors. For the latter, we employ a HyperNet (Ha et al., 2017) that learns to map directly from the binary architecture encoding to the weight space.

We hypothesize that so long as the HyperNet learns to generate reasonable weights, the validation error of networks with generated weights will correlate with the performance when using normally trained weights, with the difference in architecture being the primary factor of variation. Throughout the paper, we refer to the entire apparatus during the first part of training (the HyperNet, the variable architecture main network, and any freely learned main network weights) as the SMASH network, and we refer to networks trained with freely learned weights in the second stage as resulting networks.

## 3.1 DEFINING VARIABLE NETWORK CONFIGURATIONS

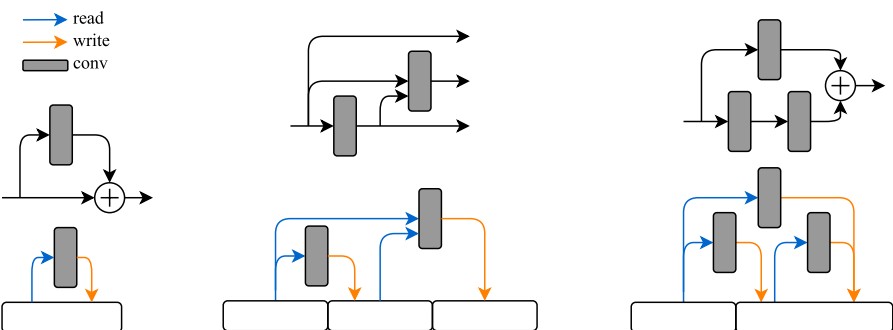

Figure 1: Memory-Bank representations of ResNet, DenseNet, and FractalNet blocks.

In order to explore a broad range of architectures with variable depth, connectivity patterns, layer sizes and beyond, we require a flexible mechanism for defining such architectures, which we can also easily encode into a conditioning vector for the HyperNet. To this end, we introduce a "memory-bank" view of feed-forward networks.

Rather than viewing a network as a series of operations applied to a forward-propagating signal, we view a network as having a set of memory banks (initially tensors filled with zeros) which it can read and write. Each layer is thus an operation that reads data from a subset of memory, modifies the data, and writes the result to another subset of memory. For a single-branch architecture, the network has one large memory bank it reads and overwrites (or, for a ResNet, adds to) at each op. A branching architecture such as a DenseNet reads from all previously written banks and writes to an empty bank, and a FractalNet follows a more complex read-write pattern, as shown in Figure 1.

Our base network structure consists of multiple blocks (Figure 2(b)), where each block has a set number of memory banks at a given spatial resolution, with successively halved spatial resolutions as in most CNN architectures. Downsampling is accomplished via a 1x1 convolution followed by

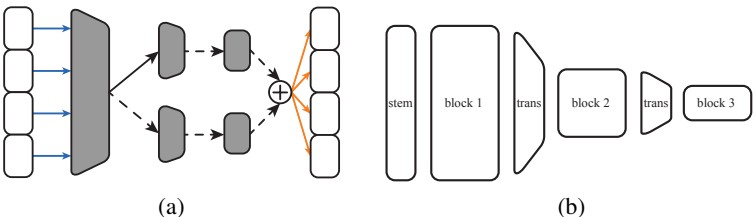

(a)                                                    (b)

Figure 2: (a) Structure of one op: A 1x1 conv operating on the memory banks, followed by up to 2 parallel paths of 2 convolutions each. (b) Basic network skeleton.

average pooling (Huang et al., 2017), with the weights of the 1x1 convolution and the fully-connected output layer being freely learned, rather than generated.

When sampling an architecture, the number of banks and the number of channels per bank are randomly sampled at each block. When defining each layer within a block, we randomly select the read-write pattern and the definition of the op to be performed on the read data. When reading from multiple banks we concatenate the read tensors along the channel axis, and when writing to banks we add to the tensors currently in each bank. For all reported experiments, we only read and write from banks at one block (i.e. one spatial resolution), although one could employ resizing to allow reading and writing from any block, similar to (Saxena and Verbeek, 2016).

Each op comprises a 1x1 convolution (reducing the number of incoming channels), followed by a variable number of convolutions interleaved with nonlinearities, as shown in Figure 2(a). We randomly select which of the four convs are active, along with their filter size, dilation factor, number of groups, and the number of output units (i.e. the layer width). The number of output channels of the 1x1 conv is some factor of the width of the op, chosen as the "bottleneck ratio" hyperparameter.

The weights for the 1x1 convolution are generated by the HyperNet as described in Section 3.2, while the other convolutions are normally learned parameters. To ensure variable depth, we learn a single set of 4 convolutions for each block, and share it across all ops within a block. We limit the max filter size and number of output units, and when a sampled op uses less than the maximum of either, we simply slice the weight tensor to the required size. The fixed transition convolutions and output layer employ this same slicing based on the number of incoming non-empty memory banks. Exact details regarding this scheme are available in the appendix.

In designing our scheme, we strive to minimize the number of static learned parameters, placing the majority of the network's capacity in the HyperNet. A notable consequence of this goal is that we only employ BatchNorm (Ioffe and Szegedy, 2015) at downsample layers and before the output layer, as the layer-specific running statistics are difficult to dynamically generate. We experimented with several different normalization schemes including WeightNorm (Salimans and Kingma, 2016), LayerNorm (Ba et al., 2016) and NormProp (Arpit et al., 2016) but found them to be unstable in training.

Instead, we employ a simplified version of WeightNorm where we divide the entirety of each generated 1x1 filter by its Euclidean norm (rather than normalizing each channel separately), which we find to work well for SMASH and to only result in a minor drop in accuracy when employed in fixed-architecture networks. No other convolution within an op is normalized.

## 3.2 LEARNING TO MAP ARCHITECTURES TO WEIGHTS

A HyperNet (Ha et al., 2017) is a neural net used to parameterize the weights of another network, the main network. For a Static HyperNet with parameters $H$, the main network weights $W$ are some function (e.g. a multilayer perceptron) of a learned embedding $z$, such that the number of learned weights is typically smaller than the full number of weights for the main network. For a Dynamic HyperNet, the weights $W$ are generated conditioned on the network input $x$, or, for recurrent networks, on the current input $x_t$ and the previous hidden state $h_{t-1}$.

We propose a variant of a Dynamic HyperNet which generates the weights $W$ based on a tensor encoding of the main network architecture $c$. Our goal is to learn a mapping $W = H(c)$ that is

Figure 3: An unrolled graph, its equivalent memory-bank representation, and its encoded embedding.

reasonably close to the optimal $W$ for any given $c$, such that we can rank each $c$ based on the validation error using HyperNet-generated weights. We thus adopt a scheme for the layout of $c$ to enable sampling of architectures with wildly variable topologies, compatibility with the toolbox available in standard libraries, and to make $c$'s dimensions as interpretable as possible.

Our HyperNet is fully convolutional, such that the dimensionality of the output tensor $W$ varies with the dimensionality of the input $c$, which we make a 4D tensor of the standard format Batch x Channel x Height x Width; the batch size is set to 1 so that no output elements are wholly independent. This allows us to vary the depth and width of the main network by increasing the height or width of $c$. Under this scheme, every slice of the spatial dimensions of $W$ corresponds to a specific subset of $c$. Information describing the op that uses that $W$ subset is embedded in the channel dimension of the corresponding $c$ slice.

For example, if an op reads from memory banks 1, 2, and 4, then writes to 2 and 4, then the first, second, and fourth channels of the corresponding slice of $c$ will be filled with 1s (indicating the read pattern) and the sixth and eighth channels of that slice will be filled with 1s (indicating the write pattern). The rest of the op description is encoded in the remaining channels in a similar 1-hot fashion. We only encode into the width-wise extent of $c$ based on the number of output units of the op, so elements of $c$ which do not correspond to any elements of $W$ are empty.

A naïve implementation of this scheme might require the size of $c$ to be equal to the size of $W$, or have the HyperNet employ spatial upsampling to produce more elements. We found these choices poor, and instead employ a channel-based weight-compression scheme that reduces the size of $c$ and keeps the representational power of the HyperNet proportional to that of the main networks. We make the spatial extent of $c$ some fraction $k$ of the size of $W$, and place $k$ units at the output of the HyperNet. We then reshape the resulting $1 \times k \times height \times width$ tensor to the required size of $W$. $k$ is chosen to be $DN^2$, where $N$ is the minimum memory bank size, and $D$ is a "depth compression" hyperparameter that represents how many slices of $W$ correspond to a single slice of $c$. Complete details regarding this scheme (and the rest of the encoding strategy) are available in Appendix B.

## 4 EXPERIMENTS

We apply SMASH to several datasets, both for the purposes of benchmarking against other techniques, and to investigate the behavior of SMASH networks. Principally, we are interested in determining whether the validation error of a network using SMASH-generated weights (the "SMASH score") correlates with the validation of a normally trained network, and if so, the conditions under which the correlation holds. We are also interested in the transferability of the learned architectures to new datasets and domains, and how this relates to normal (weight-wise) transfer learning.

Our publicly available code[1] is written in PyTorch (Paszke et al., 2017) to leverage dynamic graphs, and explicitly defines each sampled network in line with the memory-bank view to avoid obfuscating its inner workings behind (potentially more efficient) abstractions. We omit many hyperparameter details for brevity; full details are available in the appendices, along with visualizations of our best-found architectures.

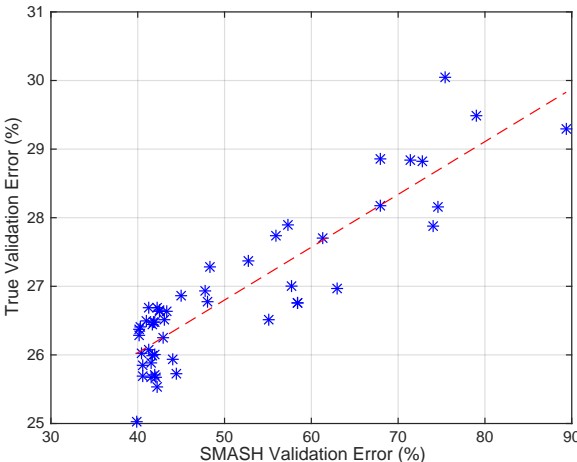

Figure 4: True error and SMASH validation error for 50 different random architectures on CIFAR-100. Red line is a least-squares best fit.

## 4.1 Testing the SMASH correlation

First, we train a SMASH network for 300 epochs on CIFAR-100, using a standard annealing schedule (Huang et al., 2017), then sample 250 random architectures and evaluate their SMASH score on a held-out validation set formed of 5,000 random examples from the original training set. We then sort the architectures by their SMASH score and select every 5th architecture for full training and evaluation, using an accelerated training schedule of 30 epochs. For these networks, which we deem SMASHv1, the architecture uses a fixed memory bank size (though a variable number of banks in each block), a single fixed 3x3 conv in the main body of the op (rather than the variable 2x2 array of convs), a single group, and a fixed bottleneck ratio of 4. The variable elements comprise the read-write pattern, the number of output units, and the dilation factor of the 3x3 filter. When sampling architectures, we allocate a random, upper-bounded compute budget to each block.

Under these conditions, we observe a correlation (Figure 4) between the SMASH score and the true validation performance, suggesting that SMASH-generated weights can be used to rapidly compare architectures. It is critical not to overstate this claim; this test is arguably a single datapoint indicating that the correlation holds in this scenario, but neither guarantees the correlation's generality nor implies the range of conditions for which it will hold. We thus conduct a more thorough investigation of this correlation.

We expect, based on preliminary experiments detailed in Appendix C, that the two key variables determining the strength of the correlation are the capacity of the HyperNet, and the ratio of HyperNet-generated weights to freely learned weights. For the former, we reason that if the HyperNet lacks representational capacity, then it will be unable to learn an acceptable mapping between architectures and weights, and the generated weights will be too far from optimal to permit ranking. For the latter, we reason that if the main network has too many freely-learned weights relative to the number of dynamically generated weights, too much of its capacity will be "static" and ill-adapted to the wide range of architectures considered.

Following these hypotheses, we repeat the first experiment multiple times, varying the architecture of the HyperNet as well as the $g$ hyperparameter, which controls the maximum layer width and consequentially the ratio of freely-learned vs dynamic weights. A higher value of $g$ corresponds to relatively fewer dynamically generated weights. We consider three different ratios and five different HyperNet configurations. For each setting of $g$ we begin by training five different SMASH networks, each employing one of the candidate configurations. After training, we sample 500 random architectures as before, and then evaluate all five SMASH scores, rank them according to their average, and then train every 10th resulting network normally. We train the resulting networks for

---

[1]https://github.com/ajbrock/SMASH

a full 100 epochs (as opposed to the previous shortened schedule) and repeat training runs with a total of 5 different random seeds. Finally, we evaluate the strength and significance of the correlation between SMASH score and resulting validation performance (averaged for each architecture across runs) using Pearson's R.

Table 1: Pearson's R ($\rho$) and p-values (p) for various settings of $g$.

| Architecture | Params | $\rho$, $g$=4 | p, $g$=4 | $\rho$, $g$=8 | p, $g$=8 | $\rho$, $g$=16 | p, $g$=16 |
|---|---|---|---|---|---|---|---|
| G=[10,10,10], D=[12, 15, 8] | 4M | -0.07 | 0.62 | 0.25 | 0.09 | 0.02 | 0.9 |
| G=[20,20,20], D=[8, 10, 4] | 5M | 0.15 | 0.30 | 0.20 | 0.17 | -0.09 | 0.53 |
| G=[20,20,20], D=[12, 15, 8] | 7M | 0.31 | 0.02 | 0.24 | 0.09 | 0.05 | 0.76 |
| G=[25,25,25], D=[16, 16, 16] | 12M | 0.38 | 0.06 | 0.24 | 0.09 | 0.04 | 0.77 |
| G=[25,25,25], D=[25,25,25] | 24M | 0.04 | 0.77 | 0.20 | 0.17 | 0.11 | 0.45 |

The results of this study are reported in Table 1. The first column details the choices of growth rate and depth for each of the three blocks of the HyperNet, and the second column the number of parameters for each architecture. The values of $g$=4, $g$=8, and $g$=16 roughly correspond to average freely-learned vs dynamic weight ratios of 1:4, 1:2, and 2:1, though these vary somewhat with individual sampled architectures.

Several trends are visible in this table. First, for $g$=4, we note that the strength of correlation increases with increased HyperNet capacity up to the fourth architecture with 12M parameters, but the correlation breaks down for the largest architecture. This suggests that the strength of correlation can indeed depend on the capacity of the HyperNet, but also that the HyperNet is either potentially susceptible to overfitting or that too large a HyperNet becomes too difficult to train and produces poor weights. Second, for $g$=8, we note that the correlation varies little with changes in architecture, but is weaker than the best correlations from $g = 4$. This suggests that for a middling ratio, the capacity of the HyperNet has less effect on the correlation as there are fewer weights for it to adapt, but the strength of the correlation is weaker as each set of sampled weights is consequentially less optimal than for $g = 4$. Third, we note a complete breakdown of the correlation for $g$=16, which is in line with our expectation that placing too much capacity in a single set of statically learned weights will prevent the SMASH network from properly adapting to individual architectures.

## 4.2 ARCHITECTURAL GRADIENT DESCENT BY PROXY

As an additional test of our method, we examine whether or not the HyperNet has learned to take into account the architecture definition in $c$, or whether it ignores $c$ and naively generates an unconditional subspace of weights that happen to work well. We "trick" the HyperNet by sampling one architecture, but asking it to generate the weights for a different architecture by corrupting the encoding tensor $c$ (e.g. by shuffling the dilation values). For a given architecture, we find that SMASH validation performance is consistently highest when using the correct encoding tensor, suggesting that the HyperNet has indeed learned a passable mapping from architecture to weights.

Following this, we posit that if the HyperNet learns a meaningful mapping $W = H(c)$, then the classification error $E = f(W, x) = f(H(c), x)$ can be backpropagated to find $\frac{dE}{dc}$, providing an approximate measure of the error with respect to the architecture itself. If this holds true, then perturbing the architecture according to the $\frac{dE}{dc}$ vector (within the constraints of our scheme) should allow us to guide the architecture search through a gradient descent-like procedure. Our preliminary tests with this idea did not yield better SMASH scores than randomly perturbing the architectural definition, though we suspect that this was in part due to our lack of an intuitively satisfying update rule for the discrete architecture space.

## 4.3 TRANSFER LEARNING

Models with weights initially learned on one large dataset frequently outperform models trained from scratch on a smaller dataset; it follows that architectures might display the same behavior. We test on STL-10 (Coates et al., 2011), a small dataset of 96x96 images similar to the CIFAR datasets. We compare the performance of the best-found architecture from CIFAR-100 (with weights trained from scratch on STL-10) to the best-found architecture from running SMASH on STL-10, and a

WRN baseline. For these experiments, we make use of the full 5,000 images in the training set; in the following section we also include comparisons against a WRN baseline using the recommended 10-fold training split.

In this case, we find that the best-found architecture from CIFAR-100 outperforms the best-found architecture from STL-10, achieving 17.54% and 20.275% error, respectively. For reference, a baseline WRN28-10 and WRN40-4 achieve respective 15.43% and 16.06% errors. This presents an interesting phenomenon: one the one hand, one might expect the architecture discovered on STL-10 to be better-tuned to STL-10 because it was specifically learned on that dataset. On the other hand, CIFAR-100 has significantly more training examples, potentially making it a better dataset for distinguishing between good architectures, i.e. accuracy on CIFAR-100 is more indicative of generality. The better performance of the architecture found on CIFAR-100 would seem to favor the latter hypothesis, suggesting that architecture search benefits from larger training sets more so than domain specificity.

We next investigate how well our best-found CIFAR-100 architecture performs on ModelNet10 (Wu et al., 2015), a 3D object classification benchmark. We train on the voxelated instances of the ModelNet10 training set using the settings of (Brock et al., 2016), and report accuracy on the ModelNet10 test set. Our 8M parameter model achieves an accuracy of 93.28%, compared to a 93.61% accuracy from a hand-designed Inception-ResNet (Brock et al., 2016) with 18M parameters trained on the larger ModelNet40 dataset.

## 4.4 BENCHMARKING

We run SMASH on CIFAR-10 and 100, augmenting our search space from the initial correlation experiment to include variable filter sizes, variable groups, and the full variable op structure shown in Figure 2, and denote the resulting networks SMASHv2. We report the final test performance of the two resulting networks with the highest SMASH scores on CIFAR-10 and 100 in Table 2.

Next, we take our best-found SMASHv2 architecture from CIFAR-100 and train it on STL-10 (Coates et al., 2011) using the recommended 10-fold training splits, and ImageNet32x32 (Chrabaszcz et al., 2017). We compare against Wide ResNet baselines from our own experiments in Tables 3 and those reported by (Chrabaszcz et al., 2017) in 4. Noting the better performance of WRN40-4 on STL-10, we also train a variant of our best architecture with only a single main convolution and 3x3 filters, to comparably reduce the number of parameters.

Table 2: Error rates (%) on CIFAR-10 and CIFAR-100 with standard data augmentation (+).

| Method | Resources | Depth | Params | C10+ | C100+ |
|---|---|---|---|---|---|
| FractalNet (Larsson et al., 2017) | | 21 | 38.6M | 5.22 | 23.30 |
| with Dropout/Drop-path | | 21 | 38.6M | 4.60 | 23.73 |
| Stochastic Depth (Huang et al., 2016) | | 110 | 1.7M | 5.23 | 24.58 |
| | | 1202 | 10.2M | 4.91 | - |
| Wide ResNet | | 16 | 11.0M | 4.81 | 22.07 |
| (Zagoruyko and Komodakis., 2016) | | 28 | 36.5M | 4.17 | 20.50 |
| DenseNet-BC ($k = 24$) (Huang et al., 2017) | | 250 | 15.3M | 3.62 | 17.60 |
| DenseNet-BC ($k = 40$) | | 190 | 25.6M | 3.46 | 17.18 |
| Shake-Shake (Gastaldi, 2017) | | 26 | 26.2M | **2.86** | **15.85** |
| NASwRL(Zoph and Le, 2017) | 800 GPUs, ? days | 39 | 32.0M | 3.84 | - |
| NASNet-A(Zoph et al., 2017) | | 20 | 3.3M | 3.41 | - |
| MetaQNN (Baker et al., 2017) | 10 GPUs, 8-10 days | 9 | 11.18M | 6.92 | 27.14 |
| Large-Scale Evolution (Real et al., 2017) | 250 GPUs, 10 days | - | 5.4M | 5.40 | - |
| | | - | 40.4 M | - | 23.7 |
| CGP-CNN (Suganuma et al., 2017) | | - | 1.68M | 5.98 | - |
| SMASHv1 (ours) | | 116 | 4.6M | 5.53 | 22.07 |
| SMASHv2 (ours) | 1 GPU, 3 days | 211 | 16M | 4.03 | 20.60 |

Our SMASHv2 nets with 16M parameters achieve final test errors of 20.60% on CIFAR-100 and 4.03% on CIFAR-10. This performance is not quite on par with state-of-the-art hand-designed networks, but compares favorably to other automatic design methods that employ RL (Baker et al., 2017; Zoph and Le, 2017) or evolutionary methods (Real et al., 2017; Suganuma et al., 2017). Our networks outperform Large-Scale Evolution (Real et al., 2017) despite requiring significantly less time to discover (though not starting from trivial models) and 10 orders of magnitude less compute. Our method outperforms MetaQNN (Baker et al., 2017) but lags behind Neural Architecture Search

Table 3: Error rates (%) on STL-10.

| Model | Params | Error |
|---|---|---|
| WRN-40-4 | 8.95M | **35.02 ± 1.14** |
| WRN-28-10 | 36.5M | 36.69 ± 2.06 |
| SMASHv2 | 16.2M | 41.52 ± 2.10 |
| SMASHv2 (3x3) | 4.38M | 37.76 ± 0.58 |

Table 4: Error rates (%) on Imagenet32x32.

| Model | Params | Top-1 | Top-5 |
|---|---|---|---|
| WRN-28-2 | 1.6M | 56.92 | 30.92 |
| WRN-28-5 | 9.5M | 45.36 | 21.36 |
| WRN-28-10 | 37.1M | 40.96 | 18.87 |
| SMASHv2 | 16.2M | **38.62** | **16.33** |

(Zoph and Le, 2017), though both methods require vastly more computation time, and unlike Neural Architecture Search, we do not postprocess our discovered architectures through grid search.

## 5 FUTURE WORK

We believe this work opens a number of future research paths. The SMASH method itself has several simplistic elements that might easily be improved upon. During training, we sample each element of the configuration one-by-one, independently, and uniformly among all possible choices. A more intelligent method might employ Bayesian Optimization (Snoek et al., 2015) or HyperBand (Li et al., 2017) to guide the sampling with a principled tradeoff between exploring less-frequently sampled architectures against those which are performing well. One might employ a second parallel worker constantly evaluating validation performance throughout training to provide signal to an external optimizer, and change the optimization objective to simultaneously maximize performance while minimizing computational costs. One could also combine this technique with RL methods (Zoph and Le, 2017) and use a policy gradient to guide the sampling. Another simple technique (which our code nominally supports) is using the HyperNet-generated weights to initialize the resulting network and accelerate training, similar to Net2Net (Chen et al., 2016).

Our architecture exploration is fairly limited, and for the most part involves variable layer sizes and skip connections. One could envision a multiscale SMASH that also explores low-level design, varying things such as the activation at each layer, the order of operations, the number of convolutions in a given layer, or whether to use convolution, pooling, or more exotic blocks. One might also search over network-wide design patterns rather than randomly selecting the read-write pattern at each layer. Alternatively, one could consider varying which elements of the network are generated by the HyperNet, which are fixed learned parameters, and one might even make use of fixed unlearned parameters such as Gabor Filters or wavelets.

Our memory-bank view also opens up new possibilities for network design. Each layer's read and write operations could be designed to use a learned softmax attention mechanism, such that the read and write locations are determined dynamically at inference time. We also do not make use of memory in the traditional "memory-augmented" sense, but we could easily add in this capacity by allowing information in the memory banks to persist, rather than zeroing them at every training step. We also only explore one definition of reading and writing, and one might for example change the "write" operation to either add to, overwrite, or perhaps even multiply (a la gated networks (Srivastava et al., 2015)) the existing tensor in a given bank.

## 6 CONCLUSION

In this work, we explore a technique for accelerating architecture selection by learning a model over network parameters, conditioned on the network's parametric form. We introduce a flexible scheme for defining network connectivity patterns and generating network weights for highly variable architectures. Our results demonstrate a correlation between performance using suboptimal weights generated by the auxiliary model and performance using fully-trained weights, indicating that we can efficiently explore the architectural design space through this proxy model. Our method achieves competitive, though not state-of-the-art performance on several datasets.

ACKNOWLEDGMENTS

We would like to thank Harri Edwards, David Ha, Daniel Angelov, Emmanuel Kahembwe, and Iain Murray for their insight. This research was made possible by grants and support from Renishaw plc and the Edinburgh Centre For Robotics. The work presented herein is also partially funded under the European H2020 Programme BEACONING project, Grant Agreement nr. 687676.

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

## APPENDIX A: HYPERPARAMETERS

We briefly describe they hyperparameters used for the SMASH network in our experiments. The SMASHv1 network has memory banks with $N = 6$ channels each, a maximum of 240 memory banks per block (though on average less than half that number), and a depth compression ratio of $D = 3$. Each layer's number of units is uniformly sampled between 6 and 42 (along even multiples of 6), and its dilation factor is uniformly sampled between 1 and 3 (with 1 representing no dilation and 3 representing 2 zeros inserted between each filter). We employ a constant bottlneck ratio of 4 as in (Huang et al., 2017), so the output of the HyperNet-generated 1x1 convolution is always 4 times the number of output units. We constrain the main network to have a maximum budget of 16M parameters, though due to our sampling procedure we rarely sample networks with more than 5M parameters.

Our SMASHv2 networks have variable memory bank sizes at each blocks, which we constrain to be multiples of $N = 8$ up to $N_{max} = 64$. We sample filter sizes from [3,5,7], and sample dilation values such that the max spatial extent of a filter in any direction is 9. We sample convolutional groups as factors of the base $N$ value (so [1,2,4,8] for these networks). We put some hand-designed priors on the choice of op configuration (i.e. which convolutions are active), giving slight preference to having all four convolutions active. For SMASHv2 nets we employ a slightly more complex bottleneck ratio: the output of the 1x1 conv is equal to the number of incoming channels while that number is less than twice the number of output units, at which point it is capped (so, a maximum bottleneck ratio of 2).

Our HyperNet is a DenseNet, designed ad-hoc to resemble the DenseNets in the original paper (Huang et al., 2017) within the confines of our encoding scheme, and to have round numbers of channels. It consists of a standard (non-bottleneck) Dense Block with 8 3x3 convolutional layers and a growth rate of 10, followed by a 1x1 convolution that divides the number of channels in two, a Dense Block with 10 layers and a growth rate of 10, another compressing 1x1 convolution, a Dense Block with 4 layers and a growth rate of 10, and finally a 1x1 convolution with the designated number of output channels. We use Leaky ReLU with a negative slope of 0.02 as a defense against NaNs, as standard ReLU would obfuscate their presence when we had bugs in our early code revisions; we have not experimented with other activations.

## APPENDIX B: ENCODING SCHEME DETAILS

We adopt a scheme for the layout of the embedding tensor to facilitate flexibility, compatibility with the convolutional toolbox available in standard libraries, and to make each dimension interpretable. First, we place some constraints on the hyperparameters of the main network: each layer's number of output units must be divisible by the memory bank size $N$ and be less than $N_{max}$, and the number of input units must be divisible by $D$, where $N$ is the number of channels in each memory bank, and $N_{max}$ and $D$ are chosen by the user. Applying these constraints allows us to reduce the size of the embedding vector by $DN^2$, as we will see shortly.

The input to a standard 2D CNN is $x \in \mathbb{R}^{B \times C \times H \times L}$, where $B$, $C$, $H$, and $L$ respectively represent the Batch, Channel, Height, and Length dimensions. Our embedding tensor is $c \in \mathbb{R}^{1 \times (2M+d_{max}) \times (N_{max}/N)^2 \times n_{ch}/D}$ where $M$ is the maximum number of memory banks in a block, $d_{max}$ is the maximum kernel dilation, and $n_{ch}$ is the sum total of input channels to the 1x1 convs of the main network.

The conditional embedding $c$ is a one-hot encoding of the memory banks we read and write at each layer. It has $2M + d_{max}$ channels, where the first $M$ channels represent which banks are being read from, the next $M$ channels represent which banks are being written to, and the final $d_{max}$ channels are a one-hot encoding of the dilation factor applied to the following 3x3 convolution. The height dimension corresponds to the number of units at each layer, and the length dimension corresponds to the network depth in terms of the total number of input channels. We keep the Batch dimension at 1 so that no signals propagate wholly independently through the HyperNet. Figure 3 shows an example of a small randomly sampled network, its equivalent memory bank representation, and how the read-write pattern is encoded in $c$. The dilation encoding is omitted in Figure 3 for compactness.

Our HyperNet has $4DN^2$ output channels, such that the output of the HyperNet is $W = H(c) \in \mathbb{R}^{1 \times 4DN^2 \times (N_{max}/N)^2 \times n_{ch}/D}$, which we reshape to $W \in \mathbb{R}^{N_{max} \times 4N_{max}n_{ch} \times 1 \times 1}$. We generate the weights for the entire main network in a single pass, allowing the HyperNet to predict weights at a given layer based on weights at nearby layers. The HyperNet's receptive field represents how far up or down the network it can look to predict parameters at a given layer. As we traverse the main network, we slice $W$ along its second axis according to the number of incoming channels, and slice along the first axis according to the width of the given layer.

## APPENDIX C: EXPERIMENT DETAILS

At each training step, we sample a network architecture block-by-block, with a random (but upper bounded) computational budget allocated to each block. For SMASHv1, We use memory banks with $N = 6$ channels each, constrain the number of incoming memory banks to be a multiple of 3 ($D = 3$), and constrain the number of output units at each layer to be a multiple of 6 (with $N_{max} = 42$) for compatibility with the memory layout.

Our HyperNet is a 26 layer DenseNet, each layer of which comprises a Leaky ReLU activation followed by a 3x3 convolution with simplified WeightNorm and no biases. We do not use bottleneck blocks, dropout, or other normalizers in the HyperNet.

When sampling our SMASHv2 networks for evaluation, we first sample 500 random architectures, then select the architecture with the highest score for further evaluation. We begin by perturbing this architecture, with a 5% chance of any individual element being randomly resampled, and evaluate 100 random perturbations from this base. We then proceed with 100 perturbations in a simple Markov Chain, where we only accept an update if it has a better SMASH score on the validation set.

When training a resulting network we make all parameters freely learnable and replace simple WeightNorm with standard BatchNorm. We tentatively experimented with using SMASH generated weights to initialize a resulting net, but found standard initialization strategies to work better, either because of a yet-undiscovered bug in our code, or because of the disparity between the dynamics of the SMASH network using WeightNorm and the resulting network using BatchNorm.

In line with our claim of "one-shot" model search, we keep our exploration of the SMASH design space to a minimum. We briefly experimented with three different settings for N and D, and use a simple, ad-hoc DenseNet architecture for the HyperNet, which we do not tune. We investigated the choice of architecture while examining the SMASH correlation, but stick to the original ad-hoc design for all benchmark experiments.

When training SMASH, we use Adam (Kingma and Ba, 2014) with the initial parameters proposed by (Radford et al., 2015) When training a resulting network, we use Nesterov Momentum with an initial step size of 0.1 and a momentum value of 0.9. For all tests other than the initial SMASHv1 experiments, we employ a cosine annealing schedule (Loshchilov and Hutter, 2017) without restarts (Gastaldi, 2017).

For the CIFAR experiments, we train the SMASH network for 100 epochs and the resulting networks for 300 epochs, using a batch size of 50 on a single GPU. On ModelNet10, we train for 100 epochs. On ImageNet32x32, we train for 55 epochs. On STL-10, we train for 300 epochs when using the full training set, and 500 epochs when using the 10-fold training splits.

For ModelNet-10 tests, we employ 3x3x3 filters (rather than fully variable filter size) to enable our network to fit into memory and keep compute costs manageable, hence why our model only has 8M parameters compared to the base 16M parameters.

All of our networks are pre-activation, following the order BN-ReLU-Conv if BatchNorm is used, or ReLU-Conv if WeightNorm is used. Our code supports both pre- and post-activation, along with a variety of other options such as which hyperparameters to vary and which to keep constant.

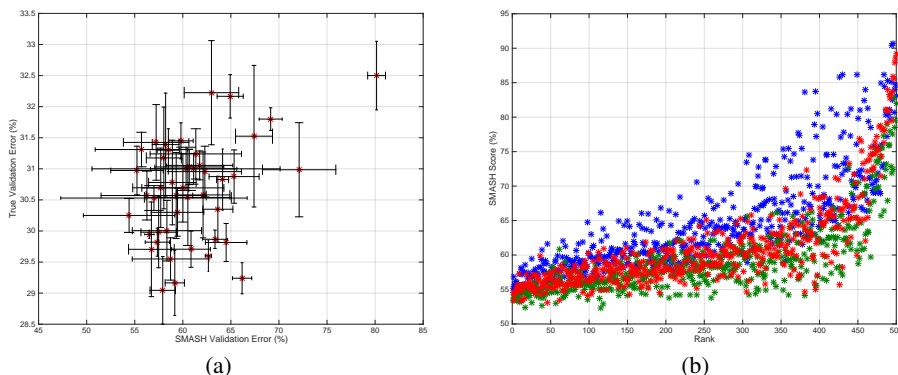

Figure 5: (a) SMASH correlation with a crippled HyperNet. Error bars represent 1 standard deviation. (b) SMASH scores vs. rank using average scores from three HyperNets with different seeds.

While investigating the SMASH correlation, we initially conducted two brief experiments to guide the choice of experiments in the remainder of the investigation. First, experiment, we train a low-budget SMASH network (to permit more rapid testing) with a much smaller HyperNet relative to the main network (though still the standard

ratio of generated to freely learned weights). We expect the decreased capacity HyperNet to be less able to learn to generate good weights for the full range of architectures, and for the correlation between SMASH score and true performance to therefore be weak or nonexistent. The results of this study are shown in Figure 5(a), where we arguably observe a breakdown of the correlation. In addition to repeat trials for each resulting net, we also train two additional SMASH networks with different random seeds and compare their predicted performance against the initial SMASH net in figure 5(b), to get a brief sense for how these values can vary across training runs.

Next, we train a high-budget SMASH network and drastically increase the ratio of normally learned parameters to HyperNet-generated parameters, such that the majority of the net's model capacity is in non-generated weights. Under these conditions, the validation errors achieved with SMASH-generated weights are much lower than validation errors achieved with an equivalent SMASH network with the typical ratio, but the resulting top models are not as performant and we found that (in the very limited number of correlation tests we performed) the SMASH score did not correlate with true performance. This highlights two potential pitfalls: first, if the HyperNet is not responsible for enough of the network capacity, then the aggregate generated and learned weights may not be sufficiently well-adapted to each sampled architecture, and therefore too far from optimal to be used in comparing architectures. Second, comparing SMASH scores for two separate SMASH networks can be misleading, as the SMASH score is a function of both the normally learned and generated weights, and a network with more fixed weights may achieve better SMASH scores even if the resulting nets are no better.

APPENDIX D: VISUALIZATIONS OF DISCOVERED ARCHITECTURES

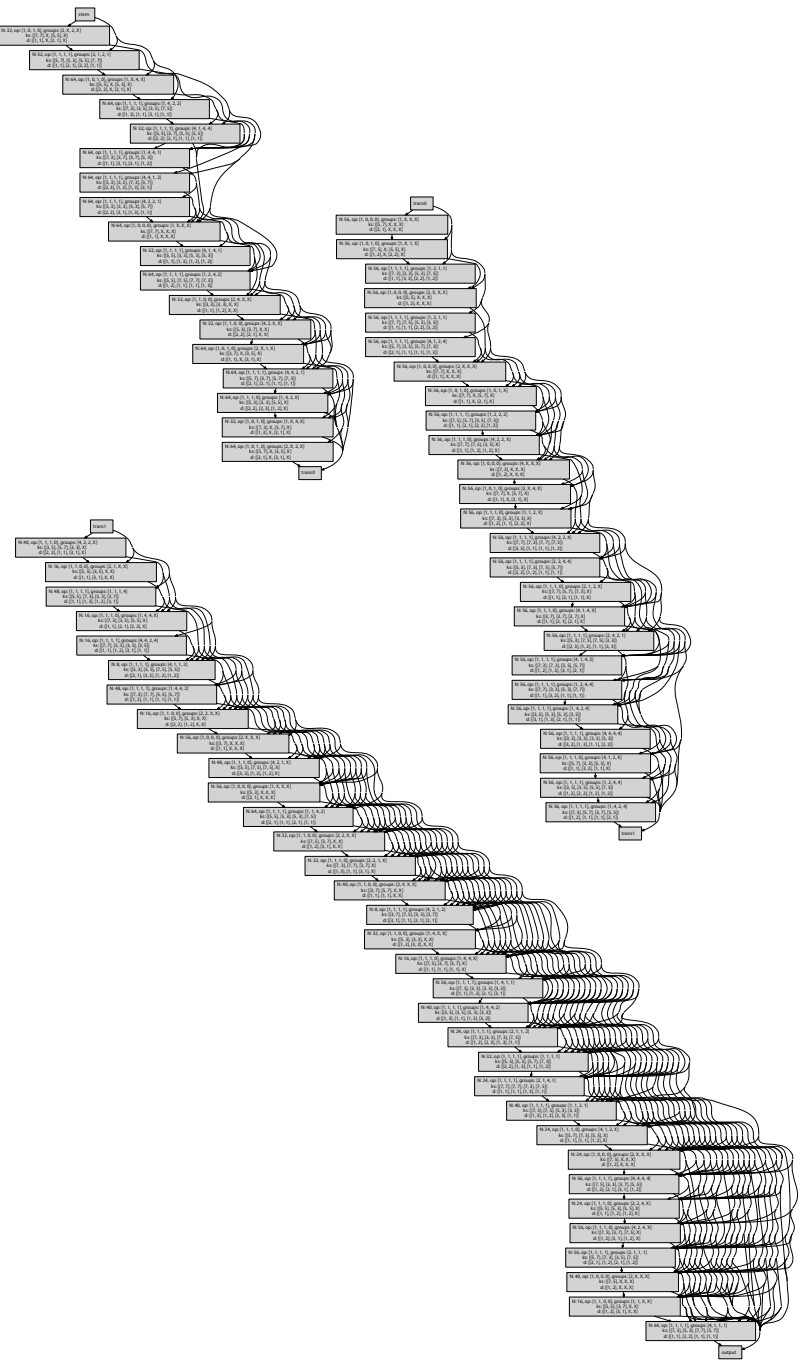

Figure 6: A simplified version of our best-found SMASHv2 architecture from CIFAR-100 with the highest SMASH score. N represents number of output units, op which convolutions are active, groups convolution groups, ks kernel size, and d dilation.

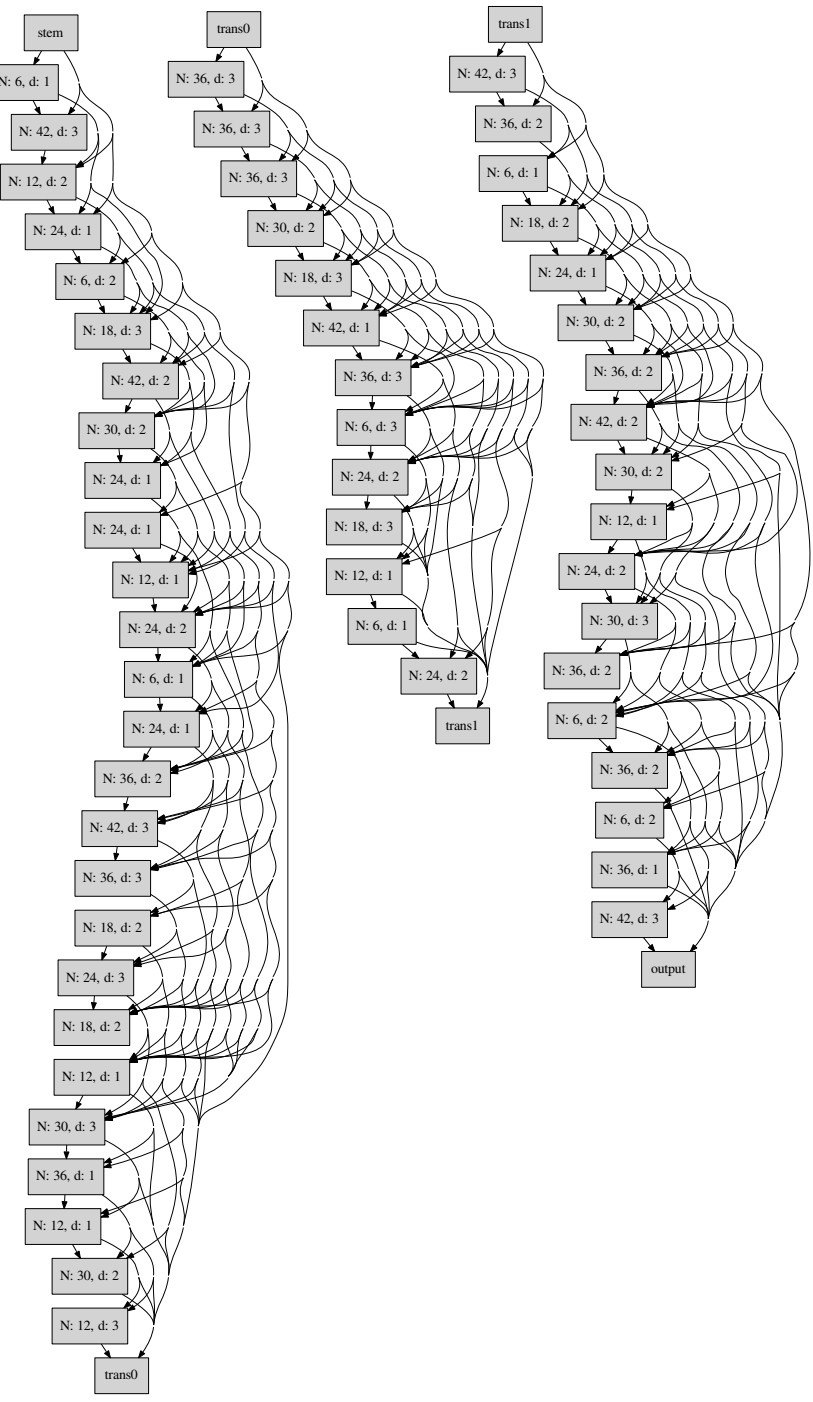

Figure 7: A simplified version of our best-found SMASHv1 architecture. N represents number of output units, and d represents the dilation factor for the 3x3 filter.

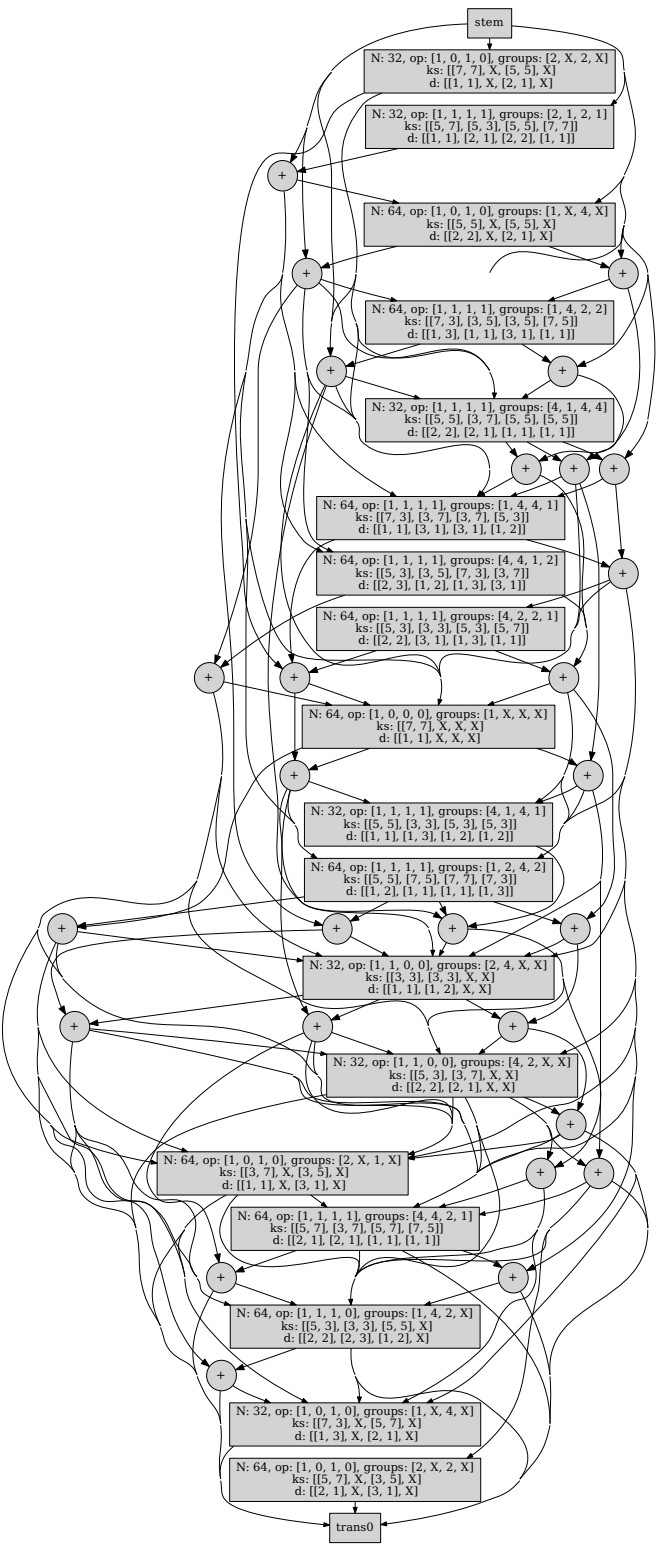

Figure 8: An expanded (though still partially simplified) view of the first block of our best SMASHv2 net.

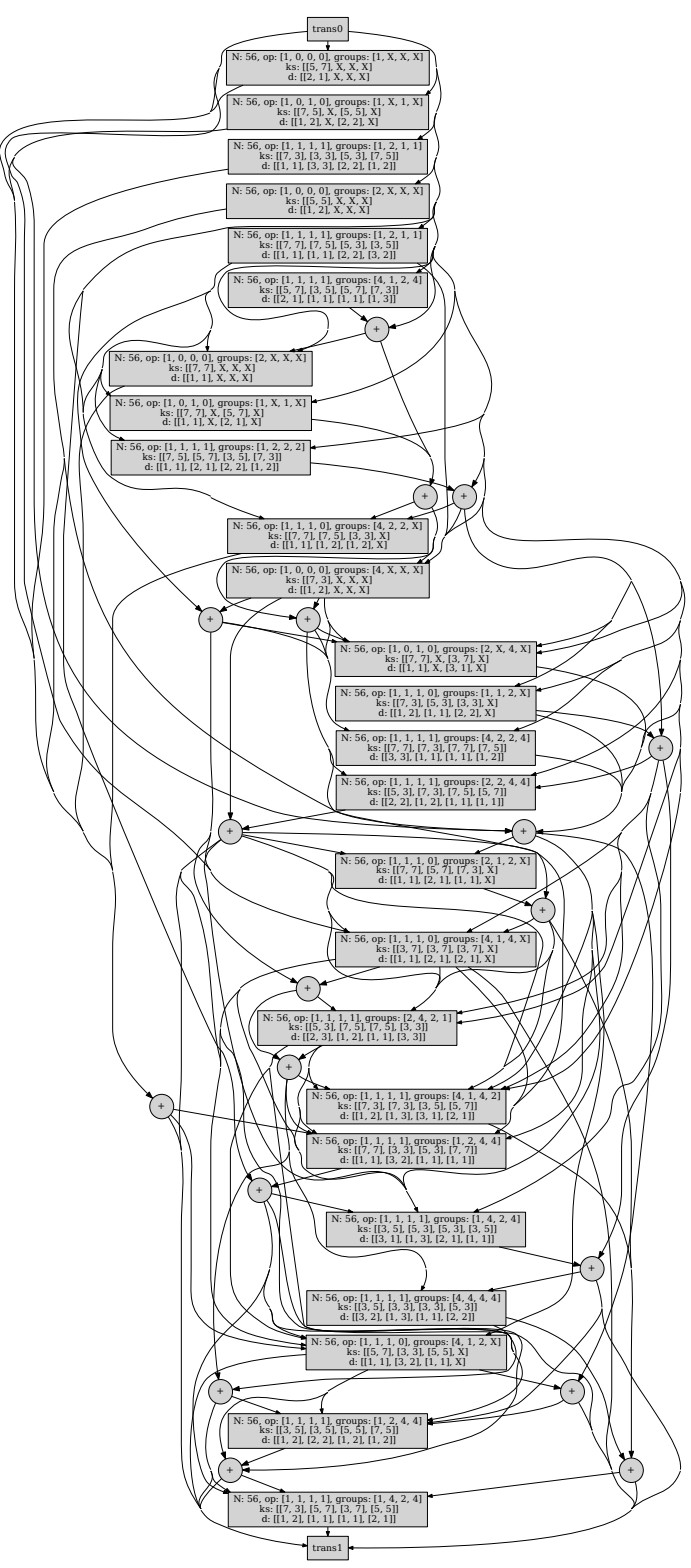

Figure 9: An expanded (though still partially simplified) view of the second block of our best SMASHv1 net.

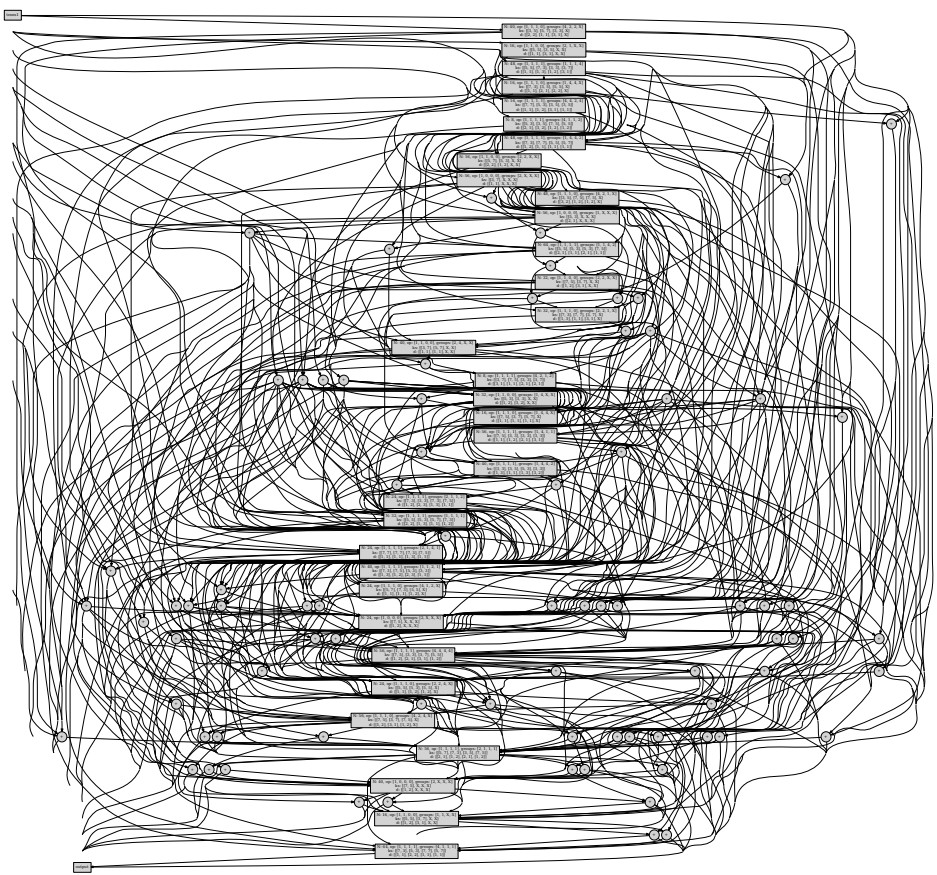

Figure 10: An expanded (though still partially simplified) view of the final block of our best SMASHv2 net. Floating paths are an artifact of the graph generation process, and are actually attached to the nearest rectangular node.

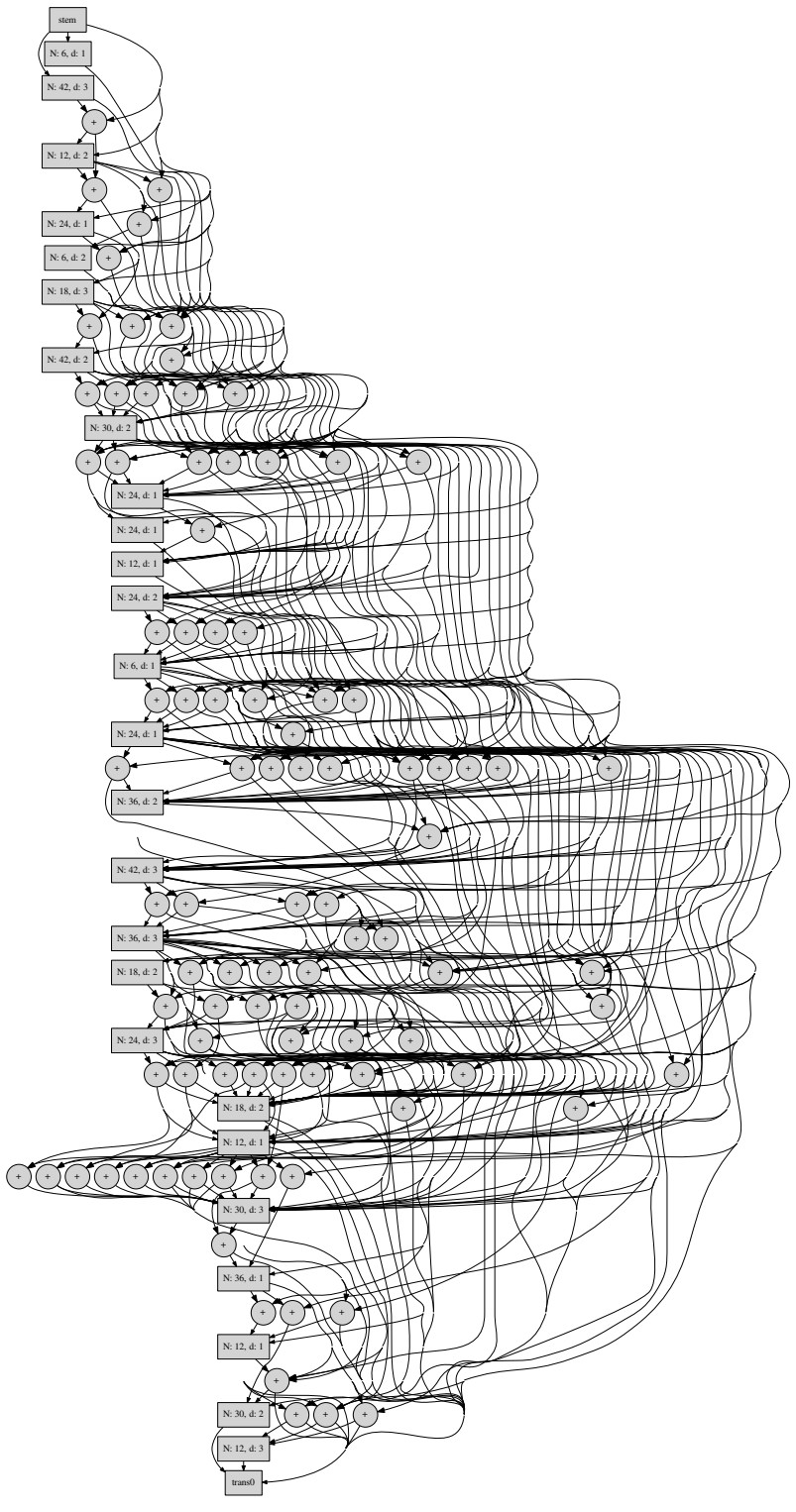

Figure 11: An expanded (though still partially simplified) view of the first block of our best SMASHv1 net. Floating paths are an artifact of the graph generation process, and are actually attached to the nearest rectangular node.

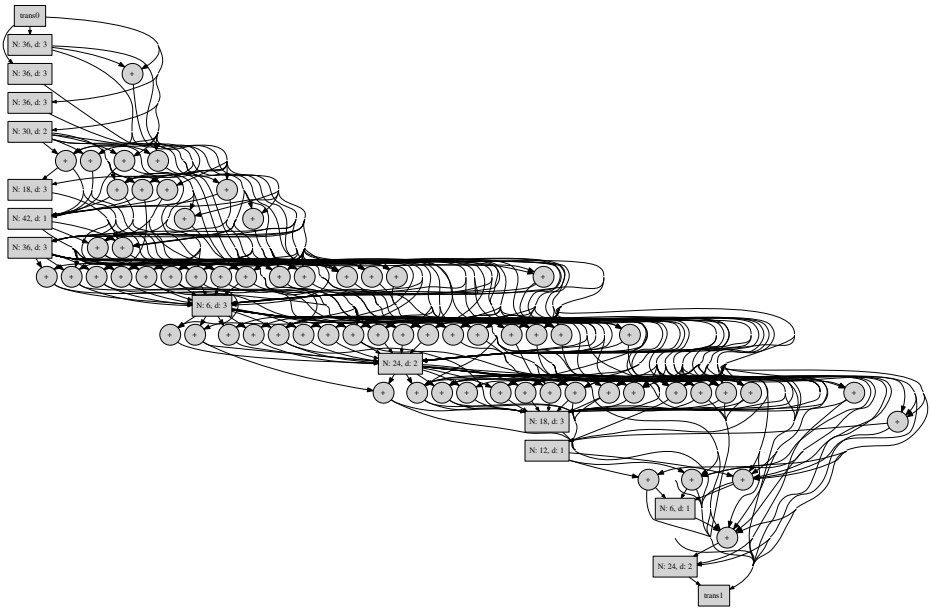

Figure 12: An expanded (though still partially simplified) view of the second block of our best SMASHv1 net. Floating paths are an artifact of the graph generation process, and are actually attached to the nearest rectangular node.

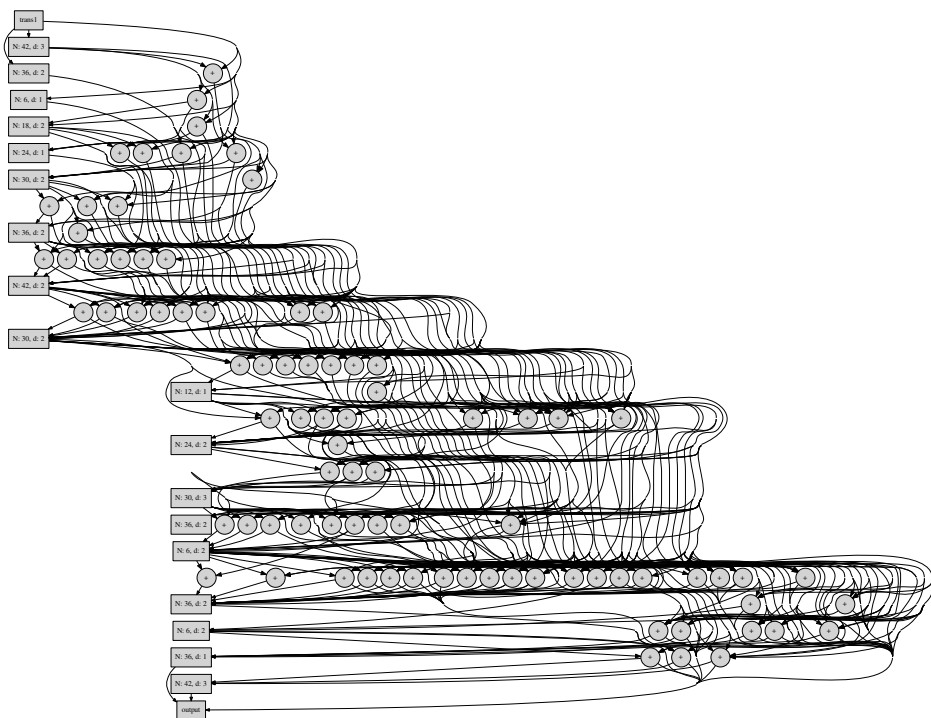

Figure 13: An expanded (though still partially simplified) view of the final block of our best SMASHv1 net. Floating paths are an artifact of the graph generation process, and are actually attached to the nearest rectangular node.

