# OpenReview forum: "SMASH: One-Shot Model Architecture Search through HyperNetworks"
_ICLR.cc/2018/Conference — Accept (Poster)_

### Official Review · AnonReviewer2 · 2017-11-26
**Well written paper that introduces and applies SMASH framework with some experimental success**

**Rating:** 7
**Confidence:** 3

**Review:**

Summary of paper - This paper presents SMASH (or the one-Shot Model Architecture Search through Hypernetworks) which has two training phases (one to quickly train a random sample of network architectures and one to train the best architecture from the first stage). The paper presents a number of interesting experiments and discussions about those experiments, but offers more exciting ideas about training neural nets than experimental successes.

Review - The paper is very well written with clear examples and an excellent contextualization of the work among current work in the field. The introduction and related work are excellently written providing both context for the paper and a preview of the rest of the paper. The clear writing make the paper easy to read, which also makes clear the various weaknesses and pitfalls of SMASH.

The SMASH framework appears to provide more interesting contributions to the theory of training Neural Nets than the application of said training. While in some experiments SMASH offers excellent results, in others the results are lackluster (which the authors admit, offering possible explanations).

It is a shame that the authors chose to push their section on future work to the appendices. The glimmers of future research directions (such as the end of the last paragraph in section 4.2) were some of the most intellectually exciting parts of the paper. This choice may be a reflection of preferring to highlight the experimental results over possible contributions to theory of neural nets.


Pros -
* Strong related work section that contextualizes this paper among current work
* Very interesting idea to more efficiently find and train best architectures
* Excellent and thought provoking discussions of middle steps and mediocre results on some experiments (i.e. last paragraph of section 4.1, and last paragraph of section 4.2)
* Publicly available code

Cons -
* Some very strong experimental results contrasted with some mediocre results
* The balance of the paper seems off, using more text on experiments than the contributions to theory.
* (Minor) - The citation style is inconsistent in places.

=-=-=-= Response to the authors

I thank the authors for their thoughtful responses and for the new draft of their paper. The new draft laid plain the contribution of the memory bank which I had missed in the first version. As expected, the addition of the future work section added further intellectual excitement to the paper.

The expansion of section 4.1 addressed and resolved my concerns about the balance of the paper by effortless intertwining theory and application. I do have one question from this section -  In table 1, the authors report p-values but fail to include them in their interpretation; what is purpose of including these p-values, especially noting that only one falls under the typical threshold for significance?

---

> ### Author Response · Authors · 2017-12-22
> **Quick Clarification**
>
> Hi Reviewer2,
>
> Thank you for your detailed and constructive feedback. We are preparing a revision and a complete response but would like a quick bit of clarification as to specifically which experimental results fall under the lackluster banner.
>
> Thanks,
>
> Paper1 Authors

---

### Official Review · AnonReviewer1 · 2017-11-27
**An experimental framework for designing neural architectures**

**Rating:** 7
**Confidence:** 4

**Review:**

This paper is about a new experimental technique for exploring different neural architectures. It is well-written in general, numerical experiments demonstrate the framework and its capabilities as well as its limitations.

A disadvantage of the approach may be that the search for architectures is random. It would be interesting to develop a framework where the search for the architecture is done with a framework where the updates to the architecture is done using a data-driven approach. Nevertheless, there are so many different neural architectures in the literature and this paper is a step towards comparing various architectures efficiently.

Minor comments:

1) Page 7,  ".. moreso than domain specificity."

It may be better to spell the word "moreso" as "more so", please see: https://en.wiktionary.org/wiki/moreso

---

### Official Review · AnonReviewer3 · 2017-12-01
**The paper tackles an important problem on learning neural net architectures that outperforms comparable methods and is reasonably faster**

**Rating:** 6
**Confidence:** 2

**Review:**

This paper tackles the problem of finding an optimal architecture for deep neural nets . They propose to solve it by training an auxiliary HyperNet to generate the main model. The authors propose the so called "SMASH" algorithm that ranks the neural net architectures based on their validation error. The authors adopt a memory-bank view of the network configurations for exploring a varied collection of network configurations. It is not clear whether this is a new contribution of this paper or whether the authors merely adopt this idea.  A clearer note on this would be welcome. My key concern is with the results as described in 4.1.; the correlation structure breaks down completely for "low-budget" SMASH in Figure 5(a) as compared Figure (4). Doesn't this then entail an investigation of what is the optimal size of the hyper network? Also I couldn't quite follow the importance of figure 5(b) - is it referenced in the text? The authors also note that SMASH is saves a lot of computation time; some time-comparison numbers would probably be more helpful to drive home the point especially when other methods out-perform SMASH.
One final point, for the uninitiated reader- sections 3.1 and 3.2 could probably be written somewhat more lucidly for better access.

---

### Author Response · Authors · 2018-01-05
**Review Response**

We would like to thank each of the reviewers for their time and constructive feedback, which we have incorporated in this revision. Specifically:

-We updated second and third parts of  Section 4.1 to more thoroughly investigate the correlation between SMASH scores and resulting validation scores by examining scores for a variety of HyperNet architectures and ratios of generated vs. static weights. We examine the strength and significance of the correlation between SMASH scores and validation scores using Pearson's R. We have moved the two original experiments addressing the breakdown of the correlation to the appendix, and updated them to properly reference the previously unreferenced figure.

-We have slightly improved writing throughout, fixing the noted typos, and changing our wording to make clear that the memory bank view is a novel development which we are introducing in this work.

-We have moved the Future Work section from the appendix into the main body of the paper at the suggestion of Reviewer 2. This section had previously been relegated to the appendix at the behest of a previous review cycle for an earlier revision of the paper.

-We have added some simple runtime numbers in Table 2 for comparison to other architecture search methods

Thanks again for your reviews.

Best,

Paper1 Authors

---

### Decision · Program_Chairs · 2018-01-29
**ICLR 2018 Conference Acceptance Decision**

**Decision:**

Accept (Poster)

**Comment:**

This paper proposes a method for having a meta deep learning model generate the weights of a main model given a proposed architecture.  This allows the authors to search over the space of architectures efficiently.  The reviewers agreed that the paper was very well composed, presents an interesting and thought provoking idea and provides compelling empirical analysis.  An exploration of the failure modes of the approach is highly appreciated.  The lowest score was also of quite low confidence, so the overall score should probably be one point higher.

Pros:
- Very well written and composed
- "Thought provoking"
- Some strong experimental results
- Analysis of weaker experimental results (failure modes)

Cons:
- Some weak results (also in pros, however)